# The Predictive Role of Raw Bioelectrical Impedance Variables in Disordered Eating Attitudes in Female Ballet Dance Students

**DOI:** 10.3390/nu12113374

**Published:** 2020-11-02

**Authors:** José Ramón Alvero-Cruz, Verónica Parent Mathias, Jerónimo C. García Romero, Ignacio Rosety, Miguel A. Rosety, Antonio Jesus Diaz, Francisco J. Ordoñez, Manuel Rosety-Rodriguez

**Affiliations:** 1Department of Human Physiology, Histology, Pathological Anatomy and Physical Education and Sport, University of Málaga-Andalucía Technology Park, 29071 Málaga, Spain; veronicaparent@hotmail.com (V.P.M.); jeronimo@uma.es (J.C.G.R.); 2The Biomedical Research Institute of Málaga (IBIMA), 29010 Málaga, Spain; 3School of Sports Medicine, Edificio López de Peñalver, Campus de Teatinos, Universidad de Málaga, 29071 Málaga, Spain; 4School of Medicine, University of Cádiz, 11003 Cádiz, Spain; ignacio.rosety@uca.es (I.R.); miguelangel.rosety@uca.es (M.A.R.); franciscojavier.ordonez@uca.es (F.J.O.); manuel.rosetyrodriguez@uca.es (M.R.-R.); 5School of Nursing, University of Cádiz, 11003 Cádiz, Spain; antoniojesus.diaz@uca.es

**Keywords:** fat mass, fat-free mass, skeletal muscle mass, phase angle, reactance, disordered eating attitudes, dance students, ROC curves

## Abstract

The present study used receiver operating characteristic (ROC) curve analysis to investigate the accuracy of body composition and raw bioelectrical impedance analysis (BIA) in correctly classifying disordered eating attitudes (DEA) in dance students. Participants were 81 female dancers assigned in two groups: beginner training (BT; age (mean ± SD) = 10.09 ± 1.2 years, *n* = 32) and advanced training (AT; age = 15.37 ± 2.1 years, *n* = 49). Fat mass (FM) was estimated by Slaughter’s equation and skeletal muscle with Poortman’s equation. Impedance (Z), resistance (R), reactance (Xc) and phase angle (PhA) were obtained through multifrequency BIA at a frequency of 50 kHz. Fat-free mass (FFM) was assessed using Sun’s equation. For evaluation of DEA, the Eating Attitudes Test-26 (EAT-26) questionnaire was performed. We defined an EAT-26 score ≥ 20 as positive for DEA. Comparisons between groups were performed by a one-way ANOVA test or Kruskall-Wallis test. Spearman’s rank correlation coefficients were performed to assess associations between variables. ROC curve analysis was utilized to test the accuracy of body composition and BIA variables in predicting DEA. In the BT group, Xc and PhA demonstrated high accuracy in predicting DEA with an area under the curve (AUC) of 0.976 (95% confidence interval (CI): 0.85–1.00) and 0.957 (95% CI: 0.82–0.99), respectively, (both *p* < 0.0001). FFM Sun had an AUC of 0.836 (95% CI: 0.66–0.94) (*p* < 0.0001) in the BT group and FFM Slaughter was 0.797 (95% CI: 0.66–0.90) (*p* < 0.001) in the AT group. Reactance and Phase angle were excellent and useful predictors of DEA in the BT group.

## 1. Introduction

Eating disorders (EDs) are psychiatric illnesses involving alterations in eating habits and their complications focus on nutritional and body composition disorders [1]. These EDs range from non-clinical eating disorders such as body image disorders [2] to severe clinical forms such as bulimia and anorexia nervosa [3]. EDs are most common in those aged 25 or younger and occur primarily among women and specifically in risk groups such as female dancers, who exhibit high levels of sophistication and are placed in competitive environments, such as dance groups, where they may be at increased risk of EDs [4]. EDs such as anorexia nervosa and bulimia produce well-known effects on body composition, including a decrease in fat (FM), fat-free mass (FFM), total body water (TBW) [1], and finally a decrease in bone mineral density [5,6].

Studies assessing nutritional status commonly use the following instruments: a clinical interrogation physical examination, anthropometry (skinfolds, BMI, arm circumference, FM, skeletal muscle mass (SMM), etc.) and biochemical analysis (albumin, hemoglobin, transferrin) [7]. Bioelectrical impedance analysis (BIA) is a common method to assess body composition based on the relationship between total body impedance and total body water. BIA is an inexpensive method used to estimate body composition and nutritional status in both healthy and ill individuals [8]. The data estimated with BIA through equations are FFM and, by derivation, FM, TBW, and extracellular water (ECW) [9], but relationships between the biophysical parameters of BIA such as reactance (Xc) and phase angle (PhA) are currently being established in different diseases such as cancer, malnutrition mortality and physical activity [8,10,11].

The use of bioelectric BIA data has gained importance in nutritional studies, and PhA is a direct measure of BIA that is not influenced by closed equations that may affect body composition compartments [12]. PhA is estimated from resistance (R) and Xc as the arc-tangent (Xc/R × 180°/π) [12]. These relationships support the hypothesis that PhA is a measure of cell mass, nutritional risk and health [6]. PhA is a variable of interest in BIA because it is not dependent of height and weight. In addition, these measurements are related to cell membrane function and are an indicator of tissular hydration and nutritional condition [13]. PhA reflects the electrical property of the tissues, whereby low values are associated with reduced cell unity, and higher values are associated with active cell mass, indicating an adequate state of health [14,15]. PhA variability is related to cell composition and function as well as the redistribution of fluids and their changes through the interstitial spaces [6]. With regard to studies on changes in PhA in diseases such as anorexia nervosa, a decrease in these values has been detected when compared to control subjects [16].

In summary, the aim of this paper consists of establishing the associations between body composition variables with disordered eating attitudes (DEA) and their predictive capacity (anthropometric and BIA estimated body composition variables) using receiver operating characteristic (ROC) analysis curves to assess DEA in a group of ballet dance students.

## 2. Material and Methods

The study protocol was approved by the Institutional Review Board of University of Málaga (EMEFYDE 2016–011). Written informed consent was obtained from each participant, during which the principles of the Declaration of Helsinki were respected.

### 2.1. Participants

Eighty-one ballet dance students between the ages of 8 and 21 years (median 95% (CI): 13.00 (12.2–14)) were enrolled in this study from the Professional Conservatory of Granada, Spain, in courses from beginner training (BT) through advanced training (AT). The students were distributed into four disciplines: contemporary, Spanish, flamenco and classical. Participation was voluntary, and prior to the study, written informed consent was obtained. The exclusion criteria were the inability to collect the anthropometric variables and incorrectly completing the Eating Attitudes Test-26 (EAT-26).

### 2.2. Anthropometric Assessment

All anthropometric measurements were taken in fasting conditions. Weight was measured on a SECA 813 electronic scale (Hamburg, Germany) to the nearest 0.1 kg, and stature was measured using a wall-mounted SECA 216 stadiometer (Hamburg, Germany) to the nearest 0.1 cm. Skinfolds were measured at: triceps and medial calf with a Holtain skinfold caliper (Holtain, Crymych, UK) accurate to 0.2 mm, and the means were computed for final calculations. Perimeters were measured with a Lufkin W606PM inextensible tape (Lufkin, México) accurate to 0.1 cm. BMI was calculated with an equation as follows: weight (kg)/height (m)^2^. Anthropometric measurements were taken according to the guidelines of the International Society for the Advancement of Kinanthropometry, [17]. The technical error of measurement of the level 3 anthropometrist was less than 3% for skinfolds and less than 1% for the rest of the anthropometric measurements.

### 2.3. Body Composition Procedures

#### 2.3.1. Anthropometric Estimations

FM was estimated by gender- and age-specific prediction equations [18] and SMM [19] using anthropometric variables. FFM was derived from body weight minus FM. 

#### 2.3.2. Bioelectrical Impedance Analysis (BIA)

BIA was performed with a body composition analyzer (MediSystem, Sanocare Human System, Madrid, Spain). The evaluation was undertaken in the early morning, with the subject under fasting conditions and having performed no moderate or intense physical exercise in the previous 24 h. Before performing the assessment, all participants urinated, were instructed to remove all metallic elements from their bodies and remained in a decubitus on a non-conductive table for 10 min (serving as an equilibration period) [20]. The impedance (Z), R, Xc and PhA values were obtained by placing four contact electrodes (PKR 170, Sanocare Human System, Madrid, Spain) on the back of the right hand and foot, introducing an alternating current of 800 mA and obtaining the results at a frequency of 50 kHz. The instrument was calibrated before each evaluation with known resistors.

BIA Estimations

FFM was calculated using the Sun et al. [21] equation, which is designed for a wide population and range of BMIs, and FM was obtained from body weight. Finally, TBW was calculated with the Sun equation [21].

### 2.4. Eating Behavior

Eating behavior was assessed with the EAT-26, that is a self-administered questionnaire. It has been validated for assessing symptoms, concerns and attitudes associated with abnormal eating behavior. The EAT-26 consists of 26 items forming three scales: dieting (related to the avoidance of fattening foods and the preoccupation with being thinner), bulimia and food preoccupation (involving items concerning thoughts about food and those indicating bulimia nervosa) and oral control (associated with the self-control of eating and the perceived pressure from others to gain weight). A total score equal to or greater than 20 on the questionnaire is indicative of abnormal eating behavior [22]. The EAT-26 test has been validated for the Spanish population [23,24,25].

### 2.5. Statistical Analysis

Normal distribution was performed using the Shapiro-Wilk test. Data were expressed as minimum, maximum, median and 95% CI. Comparisons between groups were tested by a one-way ANOVA test or Kruskall-Wallis (H test) when appropriate. Associations between the EAT-26 score subscales and body composition variables were assessed with Spearman’s coefficient of rank correlation (rho) for the two groups separately. The following criteria were adopted to interpret the level of correlations: *r* ≤ 0.1, trivial; 0.1 < *r* ≤ 0.3, small; 0.3 < *r* ≤ 0.5, moderate; 0.5 < *r* ≤ 0.7, large; 0.7 < *r* ≤ 0.9, very large; and *r* > 0.9, almost perfect [26]. ROC curve analysis was used to test the performance of the body composition and BIA variables in predicting DEA and to identify a cutoff point. The area under the curve (AUC) summarizes a test’s overall accuracy, or ability to distinguish cases from non-cases, based on the mean value of sensitivity for all possible values of specificity. AUC values are defined as non-informative (≤ 0.50), less accurate (0.51 to 0.70), moderately accurate (0.71 to 0.90), highly accurate (0.91 to 0.99), and perfect (1.0) [27]. The level of significance was set at *p* < 0.05. The statistical analysis was performed on MedCalc Statistical Software version 19.5.1 (MedCalc Software bvba, Ostend, Belgium).

## 3. Results

Concerning the anthropometric variables, statistically significant differences were found in age, weight, height and BMI (all *p* < 0.0001). Data on body composition, BMI, FM, FFM and TBW showed very significant differences between groups (all *p* < 0.0001). With regard to the EAT-26 subscales, only bulimia showed statistically significant differences (*p* = 0.004), and with respect to the BIA variables, significant differences were seen in R (*p* = 0.0007), Xc (*p* = 0.015) and PhA (*p* = 0.004) (Table 1).

### 3.1. Correlations in BT Group 

FFM Sun correlated inversely and moderately with all EAT-26 subscales and the total score (*p* < 0.05); however, neither FM Slaughter, nor FFM Slaughter, nor SMM correlated with the EAT-26 (*p* > 0.05). Similarly, Xc and PhA correlated with all EAT-26 subscales and the total score (*p* < 0.05). FM Sun and Slaughter correlated directly with the impedance variables (*p* < 0.05) (Table 2).

### 3.2. Correlations in AT Group

Moderate inverse correlations were found for FFM Sun with Z and R (both *p* < 0.05). FFM Sun showed a slight correlation with the oral control subscale (*p* < 0.05). FFM Slaughter showed only a low direct correlation bulimia and oral control (*p* < 0.05). SMM had a low correlation with oral control (*p* < 0.05). None of the BIA variables correlated with the EAT-26 (*p <* 0.05) (Table 3).

### 3.3. ROC Curve Analysis of Body Composition Variables

Of particular note are FFM Sun in kg, which had an AUC of 0.836 (*p* < 0.0001), with moderate accuracy for the BT group, and FFM Slaughter in kg with an AUC of 0.796 (*p* = 0.001), with moderate accuracy also for the AT group. The rest of the variables and anthropometric indexes including BMI and FM by BIA (Sun), FM in kg, derived by anthropometry (Slaughter) and SMM for both groups, had ROC curves with low accuracy (*p* > 0.05) (Table 4).

Sensitivity was maximal (100%) for FFM Sun in the BT group and FFM Slaughter in the AT group, and specificity was moderate (59% and 72%, respectively) (Table 5).

### 3.4. ROC Curve Analysis of BIA Variables

Notably, the AUC for Xc was 0.976 (*p* < 0. 0001) and for PhA it was 0.957 (*p* < 0. 00001), both with high accuracy in the BT group. For the AT group, none of the AUCs of the BIA variables were significant (*p* > 0.05) (Table 6).

In the BT group, the sensitivity/specificity of Xc was very high with values of 100/95.65% and a very high positive likelihood ratio of 23. The PhA also had high sensitivity/specificity values: 100/91.3% and a positive likelihood ratio of 11.5 (Table 7). The latter results have been also represented in Figure 1.

## 4. Discussion

The objective of this study was the analysis of sensitivity and specificity to discriminate DEA from body composition estimates obtained by anthropometry (FM Slaughter and derived FFM and Poortman’s SMM) as well as bioelectric variables (Z, R, Xc and PhA) and their estimates (FFM Sun and derived FM) in relation to the evaluation of their capacity in the identification of DEA. 

The novel feature of this study is the relationship between the parameters obtained from bioelectrical impedance including Xc and PhA as excellent predictors of eating disorders, especially in the BT group in comparison to the AT group.

Raw variables (Z, R, Xc and PhA) are increasingly being used in the study of body composition [6,16,28], nutrition [29,30,31], physical activity and training [11,32,33,34,35], and in the area of health and disease [8,14], etc., because they are not influenced by prediction equations, nor by BMI, FM or FFM as possible confounding variables [13].

Body composition differences between the BT and AT groups can be attributed initially to age and to the implicit difference in the components of FM and especially in the various components of FFM [6,36]. The lower values of the parameter R in the AT group are consistent with higher values of TBW and consequently of intracellular water (ICW), which means that electricity flows more easily through the body and with a minor resistance. The higher values of Xc and PhA in the AT group may indicate lower ECW values and higher ICW values. Cellular integrity has been associated with Xc. In biological conductors, theoretically, higher values are expected in BIA measurements in healthy membranes with better integrity [28]. In recent years PhA, a raw BIA variable, has been gaining attention because it is an index of the ratio between ECW and ICW, cellular integrity and body cell mass [37]. 

The relationship between PhA and BMI has been studied by Koury et al. [34] who observed a direct association (*r* = 0.58). This direct correlation between PhA and BMI (rho = 0.41, *p* < 0.05) was only found in the BT group, noting that in the AT group, PhA had an inverse association with body weight and BMI. This would be due to the gain and increase in tissue that occurs with the biological maturation of the youngest girls [29,38]. The PhA values in the study groups are 9.6° and 10.5° for the BT and AT groups, respectively, with this difference between groups being clinically important in the female gender [39] and considering that age is the most important factor in the difference in PhA [40]. The PhA increases from the earlier years of life through adolescence (18 years), with a mean value of 6.4° for girls 10–15 years of age [39], which is lower than in our two study groups. In a comparative study by Marra et al. carried out in men, the highest mean PhA value was 7.9° in the dancers, compared to both the anorexic (5.8°) and thin subjects and the controls (6.8°), with these values being much lower than those in our study, even though their study subjects were men [41]. Another reference study reported PhA values by sex and BMI ranging between 5° and 7° [39,42], with these values somewhat low in comparison to ours. These differences may be due to the country and the amount of physical exercise performed, as well as to the use of different measurement instruments [36].

There are very few data found on the relationships between BIA variables such as PhA in relation to DEA. Only one study found a significant inverse relationship between PhA and a subscale concerning hypochondria [42]. In our study, non-significant inverse correlations were found in the AT group in all the subscales of the EAT-26, while in the BT group the correlations were direct and significant. It has been proven that parameters such as PhA are sensitive after a period of refeeding in anorexic patients with protein-energy malnutrition, correlating with changes in body weight and BMI [43]. In another study, data on Xc and PhA are presented only from the perspective of body composition and nutritional changes and their physiological interpretation in anorexic patients [44].

FM showed different relationships with Xc and PhA between groups, with inverse relationships in the AT group. This finding is in line with the results of Baumgartner [6]. The associations between PhA and FM are due to and may reflect changes in TBW between FM and FFM associated with an increase in adiposity. The percentage of TBW decreases in FFM and in FM the ECW increases with increasing adiposity [6]. 

The relation between PhA and resistance is proportionally inverse, which depends on ICW and ECW. Physically working out, especially causing a muscle mass increase, may lead to an intensification in ICW, which again reduces resistance and consequently leads to a rise in PhA. Xc is directly proportional to PhA and depends on the wholeness of the cell membrane. Well-performed workouts can be a factor in improving cell membrane wholeness through the overcompensation mechanism described earlier. Another component that increases Xc is total cell mass. Workouts can lead to a rise in total cell mass that leads to an intensification in reactance and a consequent rise in PhA. Measurement of the PhA can consequently be an indicator of the effects of training on cellular health and thus on the healthiness of the person. Other authors argue that PhA might be used in clinical practice as a guide to the level of physical activity of the person [45,46].

PhA is one of the direct magnitudes obtained by BIA that does not require body weight and height determinations, and it seems to be an objective variable that is a rapid, easy and non-invasive way to deliver information about the nutritional level of participants. The biological meaning of PhA is not fully understood, but it is considered a cellular health benchmark, thus higher PhA values indicate higher cell function. Since PhA is affected by the ratio of intra- to extracellular water, it is accepted that the low values observed in older subjects reflect a decreased skeletal mass and that consequently ICW may be affected by extracellular edema/accumulation with ageing and deficient health. This suggests that PhA is an effective gauge of qualitative changes in body composition and can discriminate between levels of malnutrition. PhA describes the switch between current and voltage delivered, which is believed to be ‘in vivo’ due to the capacitance of tissue interfaces and cell membrane. Therefore, it must be affected by the volume of body cell mass (i.e., the cell compartment where most of the metabolic processes take place), changes in the cell membranes and alterations in the extracellular solutions. 

FM does not appear to be a variable that discriminates DEA; however, FFM does discriminate with an AUC of 0.836 for FFM Sun in the BT group and with an AUC of 0.79 for FFM Slaughter in the AT group, both of which are significant. SMM does not seem to be a good discriminating variable either, although it is very similar to FFM. Nonetheless, in a parallel study, the mesomorphic parameter, which indicates the musculoskeletal development of individuals, was an excellent parameter for the discrimination of DEA [47], also in the BT group.

The AUC values of Xc and PhA were very high, with a sensitivity and specificity to discriminate DEA, but only in the BT group. No similar work was found in the literature for comparison.

Several factors are suggested to contribute to developing DEA, such female gender, being overweight, living in metropolitan areas, misrepresented perception of body weight and body image dissatisfaction [48,49].

The ideal social climate for the development of DEA is probably due to several factors, such as the countless hours spent practicing in front of mirrors by ballet dancers, where they and others closely examine their bodies, as well as the seeking of a perfect body shape and perfect dance development, combined with the pressure to obtain inherent thinness in the dance and high performance expectations [50,51].

## 5. Limitations

The present study has several limitations that should be recognized. This study used a cross-sectional design that does not enable construction of a cause-effect relationship. This study was carried out on young ballet dance students restricting the generalization of these outcomes to other sports and specialties or populations.

## 6. Conclusions

Receiver operating characteristic analysis demonstrated excellent accuracy in distinguishing DEA for Xc and PhA in the BT group, better than FM, FFM or SMM by anthropometric or BIA equations. Bioelectrical PhA and Xc represent a clinically realistic procedure to assess body composition, without equation-inherent errors or needing assumptions, even if the number of body compartments is estimated. The results found in the group comprising the youngest dancers should be employed as an awareness for better control in the beginning of DEA.

## Figures and Tables

**Figure 1 nutrients-12-03374-f001:**
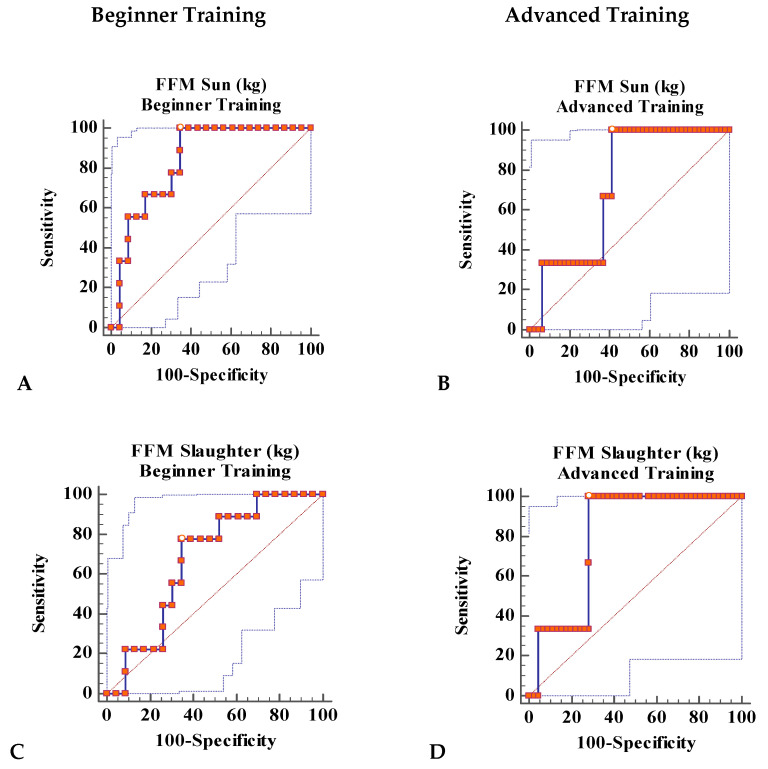
Receiver operating characteristics (ROC) curve analysis showing the area under curve (AUC) of body composition variables for the prediction of disordered eating attitudes (DEA). FFM: fat-free mass, Xc: reactance, PhA: phase angle. ROC curves are represented by the center stroke and the dotted lines are 95% confidence bounds. (**A**): FFM Sun in BT group, (**B**): FFM Sun in AT group, (**C**): FFM Slaughter in BT group, (**D**): FFM Slaughter in AT group, (**E**): Xc in BT group, (**F**): Xc in AT group (**G**): PhA BT group, (**H**): PhA in AT group.

**Table 1 nutrients-12-03374-t001:** Demographics, body composition, EAT-26 and BIA variables.

Variables	Beginner Training (*n* = 32)				Advanced Training (*n* = 49)				
	Min	Max	Median	95% CI	Min	Max	Median	95% CI	*p*
Demographics									
Age (years)	8	12	10	9.0–11.0	13	21	15	14.0–15.8	<0.0001
Weight (kg)	28.3	41.6	35.2	33.4–36.3	38.4	65.3	53.2	49.6–54.5	<0.0001
Height (cm)	119	148	141	136.9–144	147	182	158	155.1–159.0	<0.0001
Training (hours)	5	12	8	6.0–8.0	15	21	17	16.0–17.0	<0.0001
Body composition									
BMI (kg/m^2^)	15.22	21.89	17.40	16.8–18.9	16.4	23.41	21.08	19.8–21.7	<0.0001
FM Slaughter (kg)	4.723	11.57	7.47	6.8–7.9	6.50	11.86	9.07	8.4–9.9	<0.001
FFM Slaughter (kg)	22.75	34.15	27.22	25.7–27.2	31.64	55.2	43.09	41.1–44.8	<0.0001
FM Sun (kg)	0.30	9.80	4.64	3.5–7.8	2.90	17.83	13.18	11.3–14.4	<0.0001
FFM Sun (kg)	22.68	33.26	29.88	28.2–31.6	32.97	49.04	40.32	38.2–41.3	<0.0001
SMM (kg)	9.92	16.55	13.69	12.7–14.5	15.91	27.43	20.98	19.5–21.3	<0.001
TBW (kg)	18.49	26.12	23.71	22.56–25.1	25.10	38.25	31.63	30.5–32.7	<0.0001
EAT-26									
Bulimia	0	4	0	0.00–0.005	0	2	0	0.00–0.00	0.004
Oral control	0	11	2	0.0–4.0	0	11	2	1.17–3.83	0.66
Dieting	0	28	6	2–8	1	25	3	2.0–4.0	0.21
Total Score	0	40	7	4–12	2	37	5	5.0–6.0	0.46
BIA									
Z (Ω)	504	624	550.5	529–556	442	620	531	516.5–542.6	0.062
R (Ω)	496	612	545	532–553	430	609	510	499.1–527.7	0.0007
Xc (Ω)	49.2	133	88.7	85–96.0	82.4	129.1	98.3	91.5–100.5	0.015
PhA (°)	5.1	13.4	9.647	9.0–10.2	8.6	13.3	10.53	10.1–10.8	0.004

BMI: Body mass index, FM: Fat mass, FFM: Fat-free mass, SMM: Skeletal muscle mass, TBW: Total body water, BIA: Bioectrical Impedance analysis, Z: Impedance, R: Resistance, Xc: Reactance, PhA: Phase angle, EAT: Eating attitude test, CI: confidence Interval.

**Table 2 nutrients-12-03374-t002:** Spearman’s rank correlation coefficients between body composition and BIA variables with subscales EAT-26 in Basic Teachings.

	FM kg Sun	FFM kg Sun	FM kg Slaughter	FFM kg Slaughter	SMM kg	Bulimia	Oral Control	Dieting	Total Score	Z	R	Xc
FFM kg Sun	−0.12											
FM kg Slau	0.75 ***	−0.073										
FFM kg Slau	0.359 *	0.81 ***	0.233									
SMM kg	0.449 **	0.67 ***	0.46 **	0.81 ***								
Bulimia	0.16	−0.48 **	0.096	−0.257	−0.168							
Oral Control	0.135	−0.47 **	0.037	−0.321	−0.252	0.786 ***						
Dieting	0.158	−0.353 *	0.005	−0.27	−0.18	0.67 ***	0.602 **					
Total Score	0.112	−0.417 *	−0.022	−0.327	−0.251	0.77 ***	0.82 ***	0.94 ***				
Z	0.73 ***	−0.213	0.527 **	0.207	0.36 *	0.289	0.15	0.174	0.094			
R	0.75 ***	−0.237	0.646 ***	0.148	0.301	0.144	0.045	−0.088	−0.127	0.801 ***		
Xc	0.449 *	−0.287	0.453 *	0.042	0.178	0.693 ***	0.491 **	0.387 *	0.447 *	0.521 **	0.261	
PhA	0.4 *	−0.225	0.421 *	0.048	0.183	0.57 **	0.403 *	0.368 *	0.418 *	0.405 *	0.169	0.854 ***

FFM: Fat-free mass, FM: Fat mass, SMM: Skeletal muscle mass, Z: Impedance, R: Resistance, Xc: Reactance, PhA: Phase Angle, BIA: Bioectrical Impedance analysis, ** p* < 0.05, *** p* < 0.01, **** p* < 0.001.

**Table 3 nutrients-12-03374-t003:** Spearman’s rank correlation coefficients between body composition and BIA variables with subescales EAT-26 in Advanced Training Group.

	FM kg Sun	FFM kg Sun	FM kg Slaughter	FFM kg Slaughter	SMM kg	Bulimia	Oral Control	Dieting	Total Score	Z	R	Xc
FFM kg Sun	−0.023											
FFM kg Slaughter	0.603 ***	0.713 ***	0.213									
SMM kg	0.558 ***	0.616 ***	0.357 *	0.803 ***								
Bulimia	0.132	0.216	0.016	0.29 *	0.069							
Oral Control	0.291 *	0.149	0.071	0.389 **	0.288 *	0.469 **						
Dieting	−0.219	0.238	−0.122	0.01	−0.107	0.489 **	0.141					
Total Score	0.025	0.259	−0.063	0.26	0.159	0.482 **	0.828 ***	0.616 ***				
Z	0.219	−0.489 **	0.102	−0.176	−0.193	−0.034	−0.032	−0.024	−0.024			
R	0.244	−0.54 ***	0.123	−0.191	−0.186	−0.087	−0.059	−0.08	−0.081	0.94 ***		
Xc	−0.164	−0.008	0.026	−0.108	−0.094	0.103	−0.017	0.092	0.054	0.322 *	0.288 *	
PhA	−0.268	0.147	0.055	−0.14	−0.064	−0.095	−0.18	−0.034	−0.123	−0.099	−0.122	0.529 ***

FFM: Fat-free mass, FM: Fat mass, SMM: Skeletal muscle mass, Z: Impedance, R: Resistance, Xc: Reactance, PhA: Phase Angle, BIA: Bioectrical Impedance analysis, * *p* < 0.05, ** *p* < 0.01, *** *p* < 0.001.

**Table 4 nutrients-12-03374-t004:** Characteristics of receiver operating characteristics (ROC) curves of body composition variables in Beginner and Advanced groups.

Variable	Group	AUC	95% CI	*p*	Youden index J
BMI	BT	0.686	0.4–0.84	0.150	0.536
	AT	0.569	0.42–0.71	0.503	0.478
FM kg Sun	BT	0.662	0.47–0.82	0.266	0.512
	AT	0.659	0.51–0.79	0.243	0.434
FFM kg Sun	BT	0.836	0.66–0.94	<0.0001	0.652
	AT	0.717	0.57–0.84	0.073	0.587
FM kg Slaughter	BT	0.623	0.43–0.79	0.437	0.536
	AT	0.5	0.35–0.64	1	0.260
FFM kg Slaughter	BT	0.676	0.49–0.83	0.072	0.430
	AT	0.797	0.66–0.90	0.001	0.717
SMM %	BT	0.657	0.47–0.81	0.132	0.367
	AT	0.739	0.59–0.85	0.167	0.492
SMM kg	BT	0.628	0.44–0.79	0.246	0.260
	AT	0.58	0.43–0.72	0.711	0.311

BMI: Body mass index, FM: Fat mass, FFM: Fat-free mass, SMM: Skeletal muscle mass, BT: Beginner Training, AT: Advanced Training, AUC: Area under curve, CI: Confidence Interval.

**Table 5 nutrients-12-03374-t005:** Sensitivity and Specificity of body composition variables in Beginner and Advanced Training groups.

Variable	Group	Cutoff	Sensitivity	95% CI	Specificity	95% CI	+LR	95% CI	−LR	95% CI
FM kg Sun	BT	>8.341	55.56	21.2–86.3	95.65	78.1–99.9	12.78	1.7–94.8	0.46	0.2–1.0
	AT	>11.504	100	29.2–100.0	43.48	28.9–58.9	1.77	1.4–2.3	0	
FFM kg Sun	BT	≤30.136	100	66.4–100.0	65.22	42.7–83.6	2.87	1.6–5.0	0	
	AT	>40.743	100	29.2–100.0	58.7	43.2–73.0	2.42	1.7–3.4	0	
FM kg Slaughter	BT	>8.956	66.67	29.9–92.5	86.96	66.4–97.2	5.11	1.6–16.2	0.38	0.2–1.0
	AT	>8.077	100	29.2–100.0	26.09	14.3–41.1	1.35	1.1–1.6	0	
FFM kg Slaughter	BT	≤26.109	77.78	40.0–97.2	65.22	42.7–83.6	2.24	1.2–4.3	0.34	0.10–1.2
	AT	>44.990	100	29.2–100.0	71.74	56.5–84.0	3.54	2.2–5.6	0	
SMM kg	BT	≤15.327	100	66.4–100.0	26.09	10.2–48.4	1.35	1.1–1.7	0	
	AT	>24.484	33.33	0.8–90.6	97.83	88.5–99.9	15.33	1.2–189.4	0.68	0.3–1.5

FM: Fat mass, FFM: Fat-free mass, SMM: Skeletal muscle mass, BT: Beginner Training, AT: Advanced Training, CI: Confidence Interval, +LR: Positive likelihood ratio, −LR: Negative likelihood ratio.

**Table 6 nutrients-12-03374-t006:** Characteristics of receiver operating characteristics (ROC) curves of BIA variables in Beginner and Advanced groups.

Variable	Group	AUC	95% CI	*p*	Youden index J
Z	BT	0.703	0.52–0.85	0.1046	0.425
	AT	0.547	0.40–0.69	0.6317	0.391
R	BT	0.614	0.43–0.78	0.4456	0.425
	AT	0.543	0.39–0.68	0.7099	0.348
Xc	BT	0.976	0.85–1.00	<0.0001	0.956
	AT	0.598	0.45–0.73	0.5217	0.363
PhA	BT	0.957	0.82–0.99	<0.0001	0.913
	AT	0.543	0.39–0.68	0.8099	0.304

Z: Impedance, R: Resistance, Xc: Reactance, PhA: Phase angle, AUC: Area under curve, BT: Beginner Training, AT: Advanced Training, BIA: Bioectrical Impedance analysis, CI: Confidence interval.

**Table 7 nutrients-12-03374-t007:** Sensitivity and Specificity of BIA variables in Beginner and Advanced Training groups.

Variable	Group	Cutoff	Sensitivity	95% CI	Specificity	95% CI	+LR	95% CI	−LR	95% CI
Z	BT	>568	55.56	21.2–86.3	86.96	66.4–97.2	4.26	1.3–14.2	0.5	0.2–1.1
	AT	>516	100	29.2–100.0	39.13	25.1–54.6	1.64	1.3–2.1	0	
R	BT	>553	55.56	21.2–86.3	86.96	66.4–97.2	4.26	1.3–14.2	0.5	0.2–1.1
	AT	≤537	100	29.2–100.0	34.78	21.4–50.2	1.53	1.2–1.9	0	
Xc	BT	>96	100	66.4–100.0	95.65	78.1–99.9	23.0	3.4–156.4	0	
	AT	>101.9	66.67	9.4–99.2	69.57	54.2–82.3	2.19	0.9–5.5	0.5	0.10–2.4
PhA	BT	>10.1	100	66.4–100.0	91.3	72.0–98.9	11.5	3.1–43.2	0	
	AT	>9.6	100	29.2–100.0	30.43	17.7–45.8	1.44	1.2–1.7	0	

Z: Impedance, R: Resistance, Xc: Reactance, PhA: Phase Angle, BT: Beginner Training, AT: Advanced Training, CI: Confidence Interval, +LR: Positive likelihood ratio, −LR: Negative likelihood ratio.

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
