# Peer review of "The Predictive Role of Raw Bioelectrical Impedance Variables in Disordered Eating Attitudes in Female Ballet Dance Students"

_nutrients, 2020, doi:10.3390/nu12113374_

Round 1

Reviewer 1 Report

The paper presents estimates of the prediction accuracy of disordered eating attitudes (DEA) based on body composition variables 78 (anthropometric and BIA). For this purpose, the ROC curve and other measures to assess the quality of the prediction, such as sensitivity and specificity, were used.

1. Unfortunately, no reason was given as to why the correlations (Spearman's coefficient) between the variables were examined. Was the correlation study supposed to somehow bring you closer to achieving your goal? If so, how did the correlation influence the construction of the ROC curve ??? It is not clear. If the correlation is not related to the purpose of our study, then (1) these analysis should be removed from the work or (2) the purpose of the study should be extended to also include correlation analysis.

2. Tables 4 and 6 provide the Youden's J index, but the methodology lacks a description of how to interpret this index. It is not clear for what purpose this index is given here.

3. The tables should contain such measures that are then interpreted. In Tables 4 and table 6, it is not necessary to include the test statistic (z statistics), standard error (SE) values, since the confidence intervals and the p-value are provided.

4. Tables 5 and 7 present the measures describing the quality of the prediction based on the determined cut-off point. But, in the statistical methodology section authors don't specify the basis on which a given cut-off point was selected. Was it based on the Youden's J index? This should be explained in the methodology section.

5. When the obtained ROC curve does not show statistical significance, it means that it has not been proven that the given parameter that builds this curve is suitable for the forecast. This means that we will not find a good cut-off point on the ROC curve. However, researchers nevertheless designate cut-off points for ROC curves that were not statistically significant. It seems worth explaining to the readers that the points proposed at the time have no substantive significance.

6. There is no enough description of the curves in the graphs. There is a ROC curve and two other curves. These two other curves are probably the confidence interval for the ROC curve, but this is not clear.The signature under graphs should be completed.

7. What is the purpose of providing so many graphs? Many of the drawn curves are statistically insignificant and presenting them graphically does not bring any new information. I propose to leave just the charts for Xc (BT, AT), PhA (BT, AT), FFM (kg) Sun (BT, AT), FFM (kg) Slaughter (BT, AT).

8. Moreover, the number of hours of training and the age of the subjects may be a major determinant of DEA. It has not been investigated whether age and number of training hours in this study are related to Xc, PhA, FFM. It seems that such a relationship exists and that it is this relationship that may be the reason for the relationship between DEA and Xc, PhA, FFM. In the discussion, it should be noted that subsequent works should compare ROC curves based on multivariate models, e.g. based on a logistic regression model in order to eliminate the influence of age and number of hours of training and consider the independent influence of Xc, PhA, FFM on the occurrence of DEA.

Reviewer 2 Report

Alvero-Cruz et al. presented the role of raw bioelectrical impedance vaiables in disordered eating attitudes in female ballet dance students. The manuscript has the traditional composition of a scientific paper. The introduction, methods, results and conclusion are presented clearly.

However, the manuscript requires few comments:

1. The title suggest, that bioelectrical impedance variables has predictive role. In fact, none of the presented results justifies the title. According to the results, analyzed parameters revealed to be helpful in identification of DEA. Moreover, performing EAT-26 form is much easier, cheaper than performing BIA. What is clinical significance and usefulness of the findings in the context of diagnosis DEA? Have you used other diagnostic tools to diagnose DEA?

2. Please unify the rules for rounding numbers in the manuscript. Especially, there should be more numbers after the decimal point in the case of standard deviation than in the case of the mean. In my opinion there is no reason to enter some variables with such high accuracy (e.g. body weight).

3. Why have you decided to use ANOVA or Kruskal-Wallis test instead of t-test/Mann-Whitney test? What is the purpose of the comparison between beginners and advanced dancers? Are p-values really needed? Why have you used both ANOVA and Kruskal-Wallis test while only Spearman test was used for correlations? How did you choose cut-off values for DEA recognition? PPV and NPV values should be shown.

4. Did any of the participants of the study have previously diagnosed eating disorders?

5. The information about written informed consent should be following the acceptance by Ethics Committee.

6. The manuscript requires minor language correction (some sentences should be shorter)

Reviewer 3 Report

A very interesting paper.

The presented work is an attempt to assess if values of body composition parameters are related to eatig disorders in female dance students. The findings seem to be important as abnormal values of raw variables, such as reactance and phase angle were shown to be promising predictors of disordered eating attitudes that are very common in young females, especially those involved in intensive dance activities. Therefore it is important to find a „screening” tool to identify potentially compromised subjects.
Introduction to the paper presents the methods commonly used for assessment of body composition, especially variables acquired from the bioelectrical impedance method. The material and methods are described in detail. It is worth noting that results obtained in both groups, beginner training and advanced training groups, are presented clearly, however the variable FFM and FM were estimated by two different equations (Sun and Slaughter) and this fact is not mentioned in the methodology section. In the detailed statistical analysis the authors showed that the sensitivity and specificity of reactance and PhA for eating disorders were very high in beginner training dance students.
The question is why the results were specific for beginner dance students, not for advanced students. The authors tried to discuss the problem, especially in the context of factors that may influence direct variables. Altogether, the studied problem is interesting and important, but requires further investigation.
